# Modeling Obesity-Driven Pancreatic Carcinogenesis—A Review of Current In Vivo and In Vitro Models of Obesity and Pancreatic Carcinogenesis

**DOI:** 10.3390/cells11193170

**Published:** 2022-10-10

**Authors:** Sally Kfoury, Patrick Michl, Laura Roth

**Affiliations:** 1Department of Internal Medicine I, Martin-Luther University Halle/Wittenberg, Ernst-Grube-Strasse 40, D-06120 Halle (Saale), Germany; 2Department of Medicine, Internal Medicine IV, University Hospital Heidelberg, Im Neuenheimer Feld 410, D-69120 Heidelberg, Germany; 3Department of Cancer Biology, Dana-Farber Cancer Institute, Boston, MA 02215, USA; 4Department of Cell Biology, Harvard Medical School, Boston, MA 02215, USA

**Keywords:** pancreatic ductal adenocarcinoma, obesity, in vivo, in vitro

## Abstract

Pancreatic ductal adenocarcinoma (PDAC) is the most common pancreatic malignancy with a 5-year survival rate below 10%, thereby exhibiting the worst prognosis of all solid tumors. Increasing incidence together with a continued lack of targeted treatment options will cause PDAC to be the second leading cause of cancer-related deaths in the western world by 2030. Obesity belongs to the predominant risk factors for pancreatic cancer. To improve our understanding of the impact of obesity on pancreatic cancer development and progression, novel laboratory techniques have been developed. In this review, we summarize current in vitro and in vivo models of PDAC and obesity as well as an overview of a variety of models to investigate obesity-driven pancreatic carcinogenesis. We start by giving an overview on different methods to cultivate adipocytes in vitro as well as various in vivo mouse models of obesity. Moreover, established murine and human PDAC cell lines as well as organoids are summarized and the genetically engineered models of PCAC compared to xenograft models are introduced. Finally, we review published in vitro and in vivo models studying the impact of obesity on PDAC, enabling us to decipher the molecular basis of obesity-driven pancreatic carcinogenesis.

## 1. Introduction

### 1.1. Pancreatic Cancer

Pancreatic cancer is one of the most lethal cancers worldwide, associated with poor survival rates due to frequently delayed diagnosis and limited treatment options [1]. Among pancreatic cancers, pancreatic ductal adenocarcinoma (PDAC) represents the most common histological subtype, accounting for more than 90% of all cases [2]. Its incidence is dramatically increasing in the Western world, for yet widely unknown reasons. With survival rates only marginally improving, the 5-year survival rate is still appallingly low around 9% [3]. Thereby, PDAC exhibits the worst prognosis among all solid tumors [4], currently ranking as third leading cause of cancer-related deaths in the US [5]. By 2030, it is expected to become the second leading cause of cancer-related deaths in Western societies [4,6,7]. Approximately 10% of all PDAC cases are based on hereditary genetic predispositions [8]. In addition, several lifestyle factors have been shown to significantly increase the risk of developing PDAC. Besides smoking, chronic pancreatitis and diabetes mellitus, obesity represents one of the most significant risk factors [9,10].

### 1.2. Obesity

According to the World Health Organization (WHO), obesity is defined as an abnormal or excessive fat accumulation posing a substantial health risk. Obesity is usually quantified via the body mass index (BMI, defined as body mass divided by the square of the body height, expressed in units of kg/m^2^). A BMI greater than or equal to 30 is considered as obese [11]. Over the last years, the prevalence of obesity has steadily increased [12] and almost tripled since 1975 [13]. In particular, the numbers of obese children and young adults have dramatically increased during the last years, which potential aggravates the issue of obesity-related secondary diseases in the next decades [14,15]. An increased intake of high-caloric nutrition, combined with a decreased level of physical activity, are the two essential factors causing obesity in the Western world [16].

In addition to its function as crucial energy storage, adipose tissue needs to be regarded as important endocrine organ [17]. Hormones secreted from adipose tissue have been termed “adipokines”. Besides the well-known adipokine leptin, several other members such as adiponectin, resistin or visfatin belong to the adipokine family [18] and mediate systemic effects of adipose tissues. In addition to adipocytes, immune cells are the most abundant cell type within the adipose tissue [17] thereby determining its immunological impact [19]. Obesity causes a repolarization of immune cells, which is associated with a sterile inflammatory process within the adipose tissues [20,21,22,23,24], thereby inducing a systemic and chronic low-grade inflammation [25].

### 1.3. Obesity and Cancer

Overweight and obesity have previously been reported as risk factors for a variety of chronic and metabolic diseases such as type 2 diabetes mellitus, hypertension, cardiovascular disease, and metabolic syndrome [26,27]. In addition, there is a clear link between obesity and an increased risk for numerous malignancies [28,29], including pancreatic cancer [9]. Obesity is the most important avoidable risk factor for cancer [30], being responsible for 14% of cancer deaths in men and 20% of cancer deaths in women worldwide [31]. In Germany, it has been estimated that in 2018 around 7% of all newly diagnosed cancer cases were caused by obesity [32]. There is mounting evidence that both incidence and mortality of pancreatic cancer are significantly increased among obese individuals [9,33]. In line with this, it has been shown that obese people are already at a higher risk of developing pancreatic precancerous lesions [34]. The link between obesity and cancer seems to be multifactorial. In addition to the influence of proinflammatory cytokines such as IL-6 or TNF-alpha, growth-stimulating effects of various obesity-associated hormones such as leptin, estrogen, or insulin have been well described [19,35,36,37]. In terms of pancreatic cancer, coherences and mechanisms of obesity-driven carcinogenesis have been reviewed previously [38,39,40,41,42].

However, the underlying molecular mechanisms linking obesity to PDAC development and progression remain largely unknown. Therefore, it is crucial to develop realistic and physiologically accurate models of obesity-induced pancreatic carcinogenesis. This article aims to review current in vitro and in vivo models of PDAC and obesity and shed light on the newest generation of preclinical models to investigate obesity-driven pancreatic carcinogenesis (Figure 1).

Preadipocytes (blue) can be differentiated in vitro into adipocytes (yellow) with similar characteristics compared to in vivo rose adipocytes. Otherwise, mature adipocytes can be isolated out of the fat tissue and cultured for a couple of days (ceiling culture) or up to two weeks (membrane mature adipocyte aggregate cultures = MAAC) until dedifferentiation.

Pancreatic cancer cell lines (red) or isolates can be cultured in 2D and 3D models. Cancer associated fibroblasts (yellow) can be added for increased physiological relevance. Individual advantages and disadvantages are summarized in Table 3.

Common murine obesity models are based on a genetically engineered deficit in Leptin signaling (ob/ob and db/db mouse) or are the result of a high caloric diet (high fat or western diet). Pancreatic cancer in mice can arise from genetically engineered pancreas-specific mutations or induced by xenograft implantation of pancreatic cancer cells, tumor chunks, as well as organoids.

The combination of in vitro and in vivo models allows the creation of models to study obesity-driven pancreatic carcinogenesis. Individual advantages and disadvantages of the chosen models should be considered with regard to the specific scientific question. The figure was created by using BioRender (BioRender.com, accessed on 14 August 2022)

## 2. Review of Current Methodologies

### 2.1. Murine/Human Adipocyte In Vitro Models

The systemic impact of obesity is highly complex, with adipocytes interacting with multiple other cell types directly or indirectly via secreted factors [17]. In obesity-associated cancer, the crosstalk between adipocytes and immune cells is instrumental in modulating carcinogenesis and tumor progression [36]. Adipocytes account for 90% of the volume, but only for 20–40% of the total cell number in adipose tissue [17,19]. The majority of non-adipocyte cells in adipose tissues are immune cells [19]. Compared to normal-weight individuals, immune cell composition is markedly different in obese persons [20,21,25]. Considering the importance of the different immune components in obesity-driven pancreatic carcinogenesis, exploring the dynamic interaction between the adipose tissue and resident and/or infiltrating immune cells during tumor development and progression would provide further insight into the pathogenesis and possibly open new therapeutic avenues. Therefore, appropriate in vitro and in vivo models recapitulating obesity-driven pancreatic carcinogenesis and tumor progression are urgently required.

Because of their functional relevance and high prevalence in obesity, this review focuses on white adipocytes. Dufau et al. have previously published a detailed overview on different rodent and human adipose cell models [43]. Generally, in vitro differentiated adipocytes must be distinguished from isolated mature primary adipocytes (Table 1). In vitro differentiation is feasible both for murine embryonic fibroblast cell lines and primary isolated preadipocytes.

Standard mouse cell lines include 3T3-L1, 3T3-F442A, and C3H10T1/2 cells [46,47,48]. After reaching confluence, those fibroblasts can be differentiated into adipocytes by using distinct hormonal differentiation stimuli [49]. The use of cell lines offers a highly reproducible in vitro model, sparing the need to isolate primary adipose tissue. On the other hand, the cell lines used are immortalized and therefore only partly representative for primary adipocytes. In addition, several factors can influence the cell line’s capacity to differentiate in vitro, including confluence, cell passage number, serum source and lot number, contamination with mycoplasma, as well as reagent stability [48,50,51,52], creating difficulties for comparison among different labs.

For the study of primary preadipocyte cells, the most frequently used method is the isolation of stromal vascular fraction (SVF) from the rodent adipose tissues for which several protocols have been established [53]. The stromal vascular fraction contains heterogeneous cells, including adipose-derived stem cells (ADSCs), endothelial and mesenchymal progenitor cells, immune cells [44] and epithelial cells, which may limit the initial purity of the preparations. The proportion of those cell populations might vary between isolations and is affected by several factors like age, sex and nutritional stage [54,55]. In addition, different protocols used for in vitro differentiation have been shown to affect the phenotype and molecular profile of the differentiated adipocytes [51]. Nevertheless, the primary isolation of SVF from genetically modified mice enables adipocyte-specific studies on the impact of specific genetic alterations. Compared to cell line-based in vitro differentiated adipocytes, the biology and metabolism of SVF-based adipocytes are closer to that of primary mature adipocytes [56].

Compared to primary SVF preadipocytes, isolation and culture of primary mature adipocytes is experimentally challenging: these cells have a short ex vivo life span and are fragile, thereby handling can be demanding [49]. Additionally, the high lipid content causes floating of the cells, necessitating a special ceiling culture [43], for which flasks are completely filled with media and floating adipocytes attach to the upper plastic surface. A caveat of culturing mature adipocytes is their rapid dedifferentiation into fibroblast-like cells [45]. To extend the time span for culturing and decrease dedifferentiation, Harms et al. developed a new method called membrane mature adipocyte aggregate cultures (MAAC), in which mature adipocytes are cultured under a transwell membrane, thereby preventing dedifferentiation up to two weeks [45]. While this method works sufficiently for human mature adipocytes, murine mature adipocytes are even more challenging to culture [45].

Other methods to culture mature adipocytes are tissue explant cultures, which are often used to investigate adipose tissue-derived inflammation and metabolic activity [43]. As it is the case for most primary cells, these cultured adipocytes also change their phenotype after a few days ex vivo [45].

The availability of murine adipose tissue compared to human primary material is apparently much simpler, and murine adipose cell models have traditionally been most commonly utilized. However, translating results from murine-based experiments to humans also requires in vitro models using human cells. To this extent, a handful of human cell lines are available. Yet, those cell lines result from artificial immortalization or are based on pathological conditions of the donor, which might affect the generalizability of results [43]. As described for mice, adipose-derived stem cells (ASCs) can also be isolated from human adipose tissues and differentiated in vitro into adipocytes [57,58]. However, results obtained with human adipocytes might be significantly affected by interindividual differences between the different donors [57]. In addition to ASCs, isolation and culture of mature adipocytes is also feasible but underlies similar challenges as in mice [45].

Three-dimensional (3D) culture of in vitro differentiated adipocytes enables higher differentiation rates and unilocular lipid storage [43]. Disadvantages of this technique are the underlying experimental challenges as well as the higher culturing costs.

Taken together, all in vitro models have inherent limitations, most prominently the missing complex interaction of adipocytes with other organs and cell compartments. Therefore, animal models are still necessary to investigate the effects of obesity and get a better understanding of the pathological changes.

### 2.2. Murine In Vivo Obesity Models

Since mice are the most widely used in vivo models, we focus in this review on murine obesity models. In general, either genetically modified or diet-induced mouse models have been commonly used to study the impact of obesity on a broad variety of diseases. In comparison, surgical (e.g., by inducing hypothalamic lesions) or drug-induced models play a minor role [59]. Lutz et al. [59], as well as Suleiman et al. [60], reviewed different obesity mouse models in great detail. In brief, the most commonly used genetic mouse models are based on modifications in leptin, its receptor or downstream signaling (Table 2). These mice develop obesity due to increased food intake and reduced energy expenditure [61,62]. Limitations of these models are obesity-independent leptin effects on several other cell types. In particular, leptin has a significant influence on the immune response [36,63,64] which can impact the phenotype of these mouse models, especially when studying the impact of obesity on carcinogenesis [63].

Given these limitations, diet-induced obesity (DIO) mouse models are most commonly used, especially since they readily recapitulate the most common, hyperalimentation-induced cause of obesity [59]. By chronic exposure to a high-calorie diet, mice gain weight and develop obesity [67]. One limitation of these models is the fact that the various diets in use differ in their nutritional content. The most commonly applied diet is a high-fat diet, in which 20–80% of its calories are based on fatty acids [67,68,69]. Due to differences in the typical human diet in the Western world, which is predominantly carbohydrate-based, some researchers use a Western diet which more closely reflects the human dietary habits in developed Western countries [70]. A limitation of all DIO models is the uncertainty if effects are caused by obesity directly or by other factors like nutritional content or obesity-associated stress [67]. All mouse models allow the study of complex metabolic effects in vivo. Although animal models of obesity and related metabolic illnesses provide valuable insights, it must be kept in mind that their transferability to the human situation is limited due to variations in metabolism and physiology between mice and humans [57,59]. An example in this context is the basal metabolic rate, which is seven times higher in mice than in humans, which causes differences, e.g., in senescence [71].

## 3. Pancreatic Ductal Adenocarcinoma (PDAC)

PDAC is the most common histological type of pancreatic cancer [2]. Approximately 90% of all PDACs in humans are characterized by activating mutations in the proto-oncogene Kras as key driver [2,72], among them 98% exhibiting missense mutation in one of the three mutational hot-spots: glycine-12 (G12), glycine-13 (G13) or glutamine-71 (Q61), all causing a permanent activation of Kras [73]. Kras mutation is one of the earliest genetic events in PDAC carcinogenesis but is insufficient to drive PDAC development alone. Therefore, several additional genetic or epigenetic hits are required [74]. PDAC usually develops via different pancreatic precursor lesions, including mucinous cystic neoplasms (MCN), intraductal papillary mucinous neoplasms (IPMN) and pancreatic intraepithelial neoplasias (PanIN). Most PDACs develop from microscopic PanINs, which cannot be detected by conventional imaging methods [75]. Based on their histological appearance PanIN can be categorized into PanIN grades 1–3 [76,77,78], with PanIN 1 lesions already exhibiting Kras mutations [74]. During progression to invasive PDAC, additional inactivating mutations in tumor suppressors such as CDK2N2A, SMAD4 or TP53 are frequently acquired [72,74]. Suitable in vitro and in vivo models have been developed to recapitulate human pancreatic carcinogenesis and characterize the underlying molecular driver events in detail.

Several integrated genomic studies provided molecular PDAC classifications and correlated the probability of treatment response and survival to those categories [72,79,80]. Among them, the two major categories have been termed “classical epithelial” and “basal-like” (also called quasi-mesenchymal or squamous) subtypes [79,81]. Human tumors and PDAC cell lines frequently represent a heterogeneous continuum of subtypes rather than a constant state [81]. Interestingly, chemotherapy treatment may trigger shifts between subtypes [81]. Knowledge of the respective subtype of a certain cell line is important for interpreting in vitro results. For example, the most common human PDAC cell lines Panc-1 and MiaPaca2 are classified as basal-like subtypes, whereas Capan2 and HAPFII are classified as classical epithelial subtypes [81]. In humans, the molecular subtypes have gained increasing attention as predictive tools for selecting molecularly guided (neo)adjuvant or palliative treatment regimens [82].

In addition to a complex and heterogeneous genetic background, the tumor microenvironment in PDAC exerts an important, yet still controversial, impact on cancer development and chemoresistance, comprising up to 90% of the tumor volume [83,84,85,86,87]. Stromal components include immune cells, cancer-associated fibroblasts (CAF), endothelial and nerve cells as well as numerous extracellular matrix (ECM) components [85]. ECM is mainly produced by CAFs [88], but also by cancer cells themselves. Collagens, integrins, proteases, and proteoglycans are the predominant components of ECM [87]. It still remains inconclusive under which exact spatial and temporal circumstances ECM can support or suppress cancer progression [89]. Targeted depletion of ECM components has been shown to increase intratumoral chemotherapy concentrations in murine PDAC models [90]. However, in contrast to the expectations, pharmaceutical depletion of ECM has resulted in a more aggressive disease in clinical trials underlining the complexity of this interaction [89,91]. CAFs are usually derived from pancreatic stellate cells (PSC) [92]. Based on their secretory and local functions, they can be classified as myofibroblastic CAFs (myCAF) and inflammatory CAFs (iCAF). MyCAFs mediate direct juxtacrine interactions with cancer cells and therefore are frequently located in direct tumor cell contact [84]. They are characterized by a high expression level of alpha-smooth muscle actin (α-SMA) [84]. In contrast, iCAFs are spatially distant from cancer cells, but their induction depends on secreted cancer cell-derived mediators [84]. In turn, iCAFs can induce STAT3 signaling in PDAC [84] by producing pro-inflammatory cytokines, especially IL-6 [82] which is known to also cause several systemic effects of PDAC like cachexia [93] and decreased immunotherapy response [86,94].

All in all, there is a complex interaction between PDAC and its microenvironment. Relevant preclinical models and clinical trials must recapitulate this complex interplay, providing preclinical in vivo platforms to evaluate combinatorial targeting approaches of both tumor cell autonomous and non-autonomous components.

### 3.1. PDAC In Vitro Models

Numerous human and murine cell lines are available to study pancreatic carcinogenesis [95]. Most murine cell lines have been isolated from primary invasive murine PDACs [96]. Those PDACs were derived either from mouse models with a defined Kras-driven genetic background [65,66,77,97] or from chemically induced PDACs [98]. Many of the murine cell lines are not commercially available and have to be requested from the respective laboratories.

In contrast to murine PDAC, a wide variety of human PDAC cell lines are commercially available. In addition to varying age and sex of the donors, they also differ in their anatomic origin (primary tumor vs. metastasis) [95]. Among the most commonly used human PDAC cell lines, Panc1 and MiPaca2 were both isolated from primary tumors of male donors and are classified as poorly differentiated [99,100]. Because of the lack of human preinvasive PanIN cell lines, several attempts have been made to create cell lines resembling preinvasive PanIN cells, including a method using a lentiviral-based approach developed by Lee et al. [101].

Traditionally, all those cell lines are cultured in 2D. However, during the last years, several methods have been developed to establish 3D cultures of pancreatic cells, either alone or together with other cell types (Table 3):

Since most tumor tissues consist of ECM and numerous cellular stromal components with a close crosstalk between the tumor microenvironment and cancer cells, the development of co-culture models is instrumental to recapitulate the tumor–stroma interaction in vitro.

In a low-adherence environment, PDAC cell lines spontaneously form 3D structures through establishment of strong cell-cell connections instead of adherence to a plastic surface [103,104,107]; they start to produce ECM [103], allowing more physiological in vitro studies compared to 2D cultures [102].

The 3D cultures of (primary) cancer cells, together with cocultured stromal cells such as PSCs or inflammatory cells, are usually referred to as spheroids [103,104,108]. The addition of PSC causes higher secretion of ECM, and therefore, higher density of the spheroids [109], which enables a more realistic study of metabolism or response to chemotherapy [103]. Compared to 2D, culturing spheroids do not require many changes in culture conditions [104]. However, fibroblasts tend to form a core surrounded by cancer cells, which represents a rather unphysiological aggregation [102]. Organoids enable the prolonged ex vivo culture of healthy pancreatic cells [110] or PDAC cells over several passages [105]. Cryopreservation is also feasible [105]. Organoids usually require primary preinvasive pancreatic cells or invasive cancer cells, which can be of murine or human origin [104]. Human tissue samples can be obtained from surgeries, biopsies [106] or even fine needle aspirates [111]. Cells are cultured in an artificial extracellular matrix (e.g., Matrigel) containing hundreds of secreted proteins [112]. Matrigel enables the self-organization of a 3D structure that mimics physiological pancreatic histology [105]. Organoids are derived from single cells of murine and human tumors and recapitulate the physiological structure and tumor progression in vitro [104,105]. When PSCs are added to these spheroids, they differentiate into the two CAF subtypes: myCAFs and iCAFs [84,113], mimicking the human situation. Since organoids are usually derived from primary tumors or metastases, they represent suitable ex vivo models for personalized drug screening and may improve further personalized PDAC treatment strategies [106]. Disadvantages are high material costs and sophisticated, as well as time-consuming, culture methods [104,111].

An alternative is organotypic slice cultures. Slices of tumor tissues are cultured, thereby robustly recapitulating the individual tumor heterogeneity [104] while also enabling personalized drug screening [114]. However, slice culture preserves their biological characteristics only for a few days [115].

Furthermore, facilitated by novel technologies, innovative approaches like bioprinting [116] or organs on a chip have been developed recently. Bioprinting allows exact embedding of cells in gels and therefore leads to high controllability of the 3D structure, which mimics the physiological histology more accurately [117]. So far, the common use is complicated due to the limited availability of the required devices. Haque et al. developed a PDAC-based cancer-on-a-chip, which includes CAFs and macrophages recapitulating the tumor microenvironment [118]. However, so far this model is limited to cell lines, limiting its use [118]. At present, both methods seem to be promising opportunities but still need to be refined for routine use.

### 3.2. Murine PDAC In Vivo Models

Regarding murine in vivo models, we generally must differentiate between genetically engineered mouse models (GEMM) developing PDAC based on specific genetic mutations, and xenograft-based mouse models where PDAC develops after subcutaneous or orthotopic injection of tumor cells.

Regarding GEMMs, the traditional Cre/LoxP system is a powerful tool to edit mammalian gene expression and is the most commonly used site-specific recombinase system in mice [119,120]. By recognizing the specific loxP DNA segment the Cre recombinase mediates a targeted deletion of the DNA sequence between two loxP sides [121,122,123]. Thereby, the knock-out of a certain gene is possible. To control the activation of an artificially modified gene (e.g., mutated Kras), a stop codon flanked with two loxP (=lox-stop-lox; LSL) is placed in front of the gene [65,124]. Once the Cre/loxP recombinase deletes the stop codon, the expression of the modified gene is initiated. While an unspecific Cre/loxP recombinase deletes loxP floxed genes in the whole body, tissue-specific Cre/loxP is linked to a certain promotor [120]. LoxP-dependent gene editing occurs only in tissues or cells which express the relevant promotor gene. Several pancreas-specific Cre mice are commercially available. In context of PDAC, Pdx1 and p48 are the most commonly used promotors. Both result in a pancreatic deletion of loxP flanked DNA sequences [120]. Another milestone in the context of genetic mouse models was the introduction of an inducible Cre recombinase which is unable to enter the nucleus, and therefore remains inactive until a certain treatment (e.g., tamoxifen) is systemically applied [125].

Reflecting the high occurrence of Kras mutations in human PDAC [2], most GEMMs are based on a Kras mutation as key driver (Table 2). Initially developed in the Tuveson laboratory, the PDX-1-Cre;LSL-KRASG12D and p48-1-Cre;LSL-KRASG12D are the most common mouse models expressing mutant KrasG12D in a pancreas-specific manner [65]. These mice develop PanIN lesions which progress over time and can proceed to invasive PDAC [65]. Adding the inactivating TP53R175H mutation to these mouse models results in earlier and more frequent PDAC development [66]. Like human disease, metastases occur to the liver, lung, and peritoneum [66]. Less commonly used GEMMs employ other Cre promotors or genetic modifications, resulting in different behavior in terms of latency, penetrance, histological appearance and progression [126]. During recent years, tamoxifen-inducible mouse models facilitated the targeted activation of mutant pancreatic Kras in adult mice and, therefore, the possibility to study the earliest steps of PDAC carcinogenesis during adulthood, most closely resembling the human situation [127].

As an alternative to genetically engineered mouse models, various xenograft models have been used for decades. In general, the utilization of murine and human cell lines, organoids or tumor chunks for xenografting is possible. However, human material requires the use of immunocompromised mice [128,129,130], causing potential unphysiological results due to the lack of immune response during tumor progression. Still, patient-derived xenografts, being tumor samples grown subcutaneously or orthotopically in immunocompromised mice, allow experimental in vivo studies with a human tumor in situ [131].

When using murine PDAC cells or tumors for engraftment, immunocompetent mice can be used. This enables the use of mouse lines modified with specific genetic alterations of interest which can be grafted subcutaneously or orthotopically in otherwise syngeneic murine hosts [97].

In general, different injection/implantation sites and strategies are available, each carrying different advantages and disadvantages (Table 4) [97,132]. Subcutaneous injection in the flank is technically feasible without serious effort and allows an uncomplicated measurement of tumor growth [133]. However, the unphysiological localization and lack of pancreatic microenvironment might affect tumor behavior [134] and metastasis does usually not occur [135]. Moreover, intraperitoneal injection of cancer cells is technically easy to perform. In this case, peritoneal and liver metastasis can frequently be observed [134]. The most physiological site is the orthotopic injection into the pancreas, creating an endogenous pancreatic environment [97]. The surgical implantation which is most used for orthotopic implantation requires an experimenter experienced in performing laparotomy [136]. Alternatively, an ultrasound-based injection is also feasible. These are clearly less invasive, carrying less complications, but also require well-trained operators and are associated with a higher risk of mislocalization of the inoculated tumor cells compared to laparotomy [137]. Metastases at physiological sites frequently occur following orthotopic injections [97,134].

As an alternative to injections into the pancreas, direct liver injection, portal vein injection, or hemispleen injections have been used as liver metastasis models but also require a surgical procedure [97,138,139]. The technically most challenging model is the direct injection of organoids into the bile duct [140,141]. In this context, the direct injection of PanIN organoids into the bile duct has been described as an interesting tool to the study of early PDAC carcinogenesis [141].

Taken together, several murine PDAC models with distinct advantages and limitations have been developed during the last two decades, which should be selected in the context of the specific scientific question under consideration.

### 3.3. Utilization of In Vitro and In Vivo Models of Obesity Associated PDAC

Obesity affects cancer development and progression in multiple ways [17]. Overall, the interaction of several obesity-associated mediators significantly increases the risk of tumor development and systemic tumor spread [36,142]. Notably, obesity-induced systemic alterations include impaired nutritional parameters (e.g., blood glucose level or free fatty acids [143]), hormonal disorders (e.g., insulin, leptin, estrogen [24,144,145]), chronic inflammation [22,24,146] with restricted immune competence [147], as well as alterations within the tumor microenvironment including increased desmoplasia and activation of PSCs [148]. The combination of relevant in vivo and in vitro models of both obesity and PDAC enables the study of basic molecular mechanisms underlying obesity-induced pancreatic carcinogenesis.

In this context, Mendonsa et al. confirmed the expression of leptin receptors in several PDAC cell lines on protein level [149]. Furthermore, leptin treatment caused a significant increase in PDAC cell proliferation and migration [149]. By co-culture of in vitro differentiated adipocytes and (preinvasive) pancreatic cancer cell lines, Meyer et al. demonstrated an increased proliferation of PDAC and PanIN cells addicted to an adipocyte-dependent transfer of glutamine [150]. Conversely, transwell co-culture of adipocytes and PDAC cells confirmed an impaired adipocyte morphology and metabolism induced by the presence of PDAC cells [151].

Orthotopic implantation of PDAC cells, organoids, or tumor chunks in mouse models of diet-induced obesity, or genetically induced obesity, allows us to study effects of preexisting obesity on PDAC progression. After the orthotopic implantation of PanIN organoids in obese mice, Lupo et al. observed increased grades of dysplasia [152]. Moreover, obesity caused an accelerated tumor growth of implanted PDAC organoids [152]. Another way to study the effects of obesity on PDAC is to induce obesity in GEMM of PDAC. There, diet-induced obesity led to enhanced PanIN progression in a genetic PanIN mouse model (KC mice), eventually also resulting in more frequent development of invasive PDAC [153]. In line with these findings, both genetic (AAV-leptin-based) and diet-induced weight loss were able to abolish obesity-driven PDAC development [154]. For this study the authors applied the genetic obesity mouse model based on a leptin deficiency (=ob/ob). The application of adeno-associated virus-sustained leptin secretion causes a rapid weight loss of obese ob/ob-mice, which was associated with a reduction in PDAC tumor size [154].

Based on the simultaneous utilization of in vitro and in vivo models of both obesity and PDAC, new preliminary insights on obesity-driven PDAC carcinogenesis have been obtained. However, given the dramatically increasing prevalence of obesity in the Western world, it is of utmost importance to further refine our understanding of the underlying molecular mechanisms of obesity-driven pancreatic carcinogenesis, allowing the effective development of preventive strategies and identification tools for early diagnosis.

## 4. Conclusions and Future Perspectives

Obesity has been associated with a significantly increased risk of pancreatic cancer and contributes to poor prognosis and survival. Pathways and mechanisms to clarify this association are still not well understood. Reducing the prevalence of obesity should be the ultimate goal. Therefore, intensified population-wide education on a healthy lifestyle is crucial. However, it is clear that education alone will not suffice to achieve the goal of a population-wide weight optimization. Therefore, the investigation of druggable targets to reduce body weight is also necessary to reduce the risk of obesity-related diseases including pancreatic cancer. There is hope that a better understanding of the molecular interactions between obesity and PDAC development could open new avenues to improve diagnostic and therapeutic modalities in PDAC. However, reliable pre-clinical models that adequately depict the biology of the disease in patients are required to assess the effectiveness of new diagnostic and therapeutic approaches. Different PDAC-, obesity- and obesity-driven PDAC models are already available and must be selected with respect to their individual advantages and disadvantages. For early diagnosis of PDAC in obese high-risk individuals, multiparametric prediction models for obesity-driven early carcinogenesis based on transcriptomic, epigenomic, proteomic and metagenomic screening approaches may be identified in appropriate murine in vitro and in vivo models, and these must be validated in human samples ex vivo before entering clinical validation. For studying the impact of obesity, diet and physical activity as predictors of therapy response and tumor progression, a similar sequential combination of preclinical in vitro and in vivo approaches has to be utilized before validation in translational programs of clinical trials. Lastly, the therapeutic targeting of obesity-driven carcinogenesis as the ultimate goal also requires a concerted effort using multiomics screens to identify druggable targets dependent on obesity-driven key signaling cascades which provide novel avenues to tackle and improve the still dismal prognosis of pancreatic cancer.

## Figures and Tables

**Figure 1 cells-11-03170-f001:**
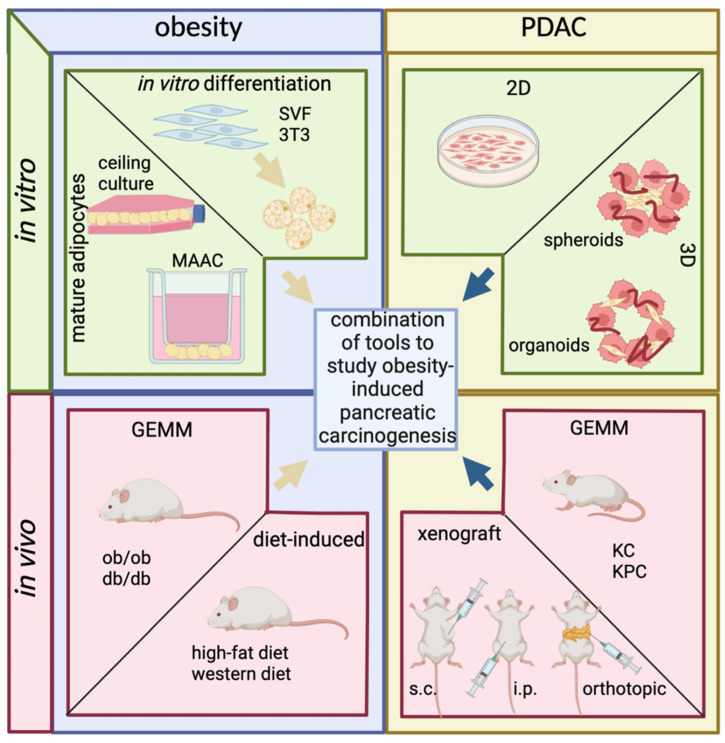
Overview of in vitro and in vivo models of obesity and PDAC.

**Table 1 cells-11-03170-t001:** Overview of different in vitro methods to culture adipocytes.

Method	Description	References
SVF	Stroma vascular fraction isolated out of the adipose tissue is differentiated in vitro into adipocytes	Kilroy et al. 2018 [44]
Ceiling culture	Mature adipocytes cultured under the upper plastic surface of a flask due to the floating characteristics of adipocytes; dedifferenciate within a few days	Dufau et al. 2021 [43]; Harms et al. 2019 [45]
mature adipocyte aggregate cultures (MAAC))	Mature adipocytes cultured under permeable small-pored membrane insert; preserves mature adipocyte identity and function for up to 14 days	Harms et al. 2019 [45]

**Table 2 cells-11-03170-t002:** Overview of the characteristics of common genetically engineered mouse models (=GEMM) of obesity and pancreatic cancer.

GEMM	Description	References
ob/ob	Leptin deficiency-induced obesity due to increased food intake and decreased energy expenditure; reversible by leptin substitution	Halaas et al. 1995 [61]
db/db	Leptin receptor defect causes obesity due to increased food intake and decreased energy expenditure; leptin substitution cannot rescue the defect	Coleman et al.1978 [62]
KC (Kras^G12D^, Pdx1 Cre)	month-long development of PanIN (all grades), in some cases development of invasive PDAC	Hingorani et al. 2003 [65]
KPC (Kras^G12D^, tp53^R175H^, Pdx1 Cre)	Rapid development of PanIN lesions and invasive PDAC with high penetrance, metastasis to the liver, lung and peritoneum	Hingorani et al. 2005 [66]

**Table 3 cells-11-03170-t003:** Common 2D and 3D cell culture methods of pancreatic cancer.

Method	Description	References
monolayer	Simple and common way to grow cell lines; less physiological because of lack of tumor microenvironment	Heinrich et al. 2021 [102]
spheroid	Spontaneous 3D formation; CAFs can be added; unphysiological configuration of PDAC and CAFs	Ware et al. 2016 [103]; Gündel et al. 2021 [104]; Lee et al. 2017 [101]; Öhlund et al. 2014 [85]
organoid	Single cell-based 3D formation with physiological structures; ambitious technique and costly	Gündel et al. 2021 [104]; Boj et al. 2015 [105]; Driehuis et al. 2019 [106]

**Table 4 cells-11-03170-t004:** Comparison of different injections in xenograft-mouse models of pancreatic cancer. Injection sides induce diverse characteristics with individual advantages and disadvantages.

Implantation Side	Description	References
subcutaneous	Allow direct observation of tumor growth; unphysiological localization; lack of pancreatic microenvironment affecting tumor behavior; no metastasis	Garrido-Laguna et al. 2011 [133]; Michaelis et al. 2017 [134]; Killion et al. 1998 [135]
intraperitoneal	Peritoneal and liver metastasis; lack of pancreatic microenvironment	Michaelis et al. 2017 [134]
pancreas	Pancreatic microenvironment present; metastasis into liver and lung; requires surgery or ultrasound-based implantation	Erstad et al. 2018 [136]
Portal vein injection/Hemispleen injection	Liver metastasis model, requires extensive surgery	Mallya et al. 2021 [97]; McVeigh et al. 2019 [138]; Au-Soares et al. 2014 [139]

## Data Availability

Not applicable.

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
