# Peer review of "Modeling Obesity-Driven Pancreatic Carcinogenesis—A Review of Current In Vivo and In Vitro Models of Obesity and Pancreatic Carcinogenesis"

_cells, 2022, doi:10.3390/cells11193170_

Round 1

Reviewer 1 Report

This is an interesting article summarising literature on in-vitro and in-vivo models of pancreatic cancer and obesity. 

The quality of the article will be improved by adding figures to explain possible concepts of obesity-induced PC.

Since obesity is a preventable risk factor, investigation of drugable targets may not be a viable and cost-effective approach.

Reviewer 2 Report

The major weakness of this manuscript is lack of novelty.

There are several review articles on association of obesity and pancreatic cancer such as

1. Bracci PM. Obesity and pancreatic cancer: overview of epidemiologic evidence and biologic mechanisms. Mol Carcinog. 2012 Jan;51(1):53-63. doi: 10.1002/mc.20778. PMID: 22162231; PMCID: PMC3348117.

2. Pothuraju R, Rachagani S, Junker WM, Chaudhary S, Saraswathi V, Kaur S, Batra SK. Pancreatic cancer associated with obesity and diabetes: an alternative approach for its targeting. J Exp Clin Cancer Res. 2018 Dec 19;37(1):319. doi: 10.1186/s13046-018-0963-4. PMID: 30567565; PMCID: PMC6299603.

Reviewer 3 Report

v

Manuscript entitled: „Modeling obesity-driven pancreatic carcinogenesis in vitro and in vivo” is a review paper regarding  problem of pancreatic cancer pathogenesis and its relation to obesity.

The paper was carefully and interestingly planed. In the Introduction authors present epidemiology of pancreatic cancer,  growing incidence of this malignancy in recent years and dramatic number of cancer-related death. The  lack of adequate treatment of this cancer results from  of inadequate knowledge  of its pathogenesis.  Next part of paper focus on the  obesity as the very serious risk factor for  the development of pancreatic tumor. The main part of the text  concerns research  studies. Authors present in vivo and in vitro  models of  pancreatic cancer and obesity, such as ; culture of adipocytes,  and pancreatic human and murine cancer cell lines, organoids, and  in vivo ; knockout mice and xenograft models.  Then  the review of publication concerning obesity -related cancerogenesis concludes that the problem is still open and needs further investigation. References are numerous (149) and appropriately cited.  

Manuscript entitled: „Modeling obesity-driven pancreatic carcinogenesis in vitro and in vivo” is a review paper regarding  problem of pancreatic cancer pathogenesis and its relation to obesity.

The paper was carefully and interestingly planed. In the Introduction authors present epidemiology of pancreatic cancer,  growing incidence of this malignancy in recent years and dramatic number of cancer-related death. The  lack of adequate treatment of this cancer results from  of inadequate knowledge  of its pathogenesis.  Next part of paper focus on the  obesity as the very serious risk factor for  the development of pancreatic tumor. The main part of the text  concerns research  studies. Authors present in vivo and in vitro  models of  pancreatic cancer and obesity, such as ; culture of adipocytes,  and pancreatic human and murine cancer cell lines, organoids, and  in vivo ; knockout mice and xenograft models.  Then  the review of publication concerning obesity -related cancerogenesis concludes that the problem is still open and needs further investigation. References are numerous (149) and appropriately cited.  

Minor suggestion;

The title is short and interesting, but because the main part of paper is related to the experimental models of pancratic cancer and obesity it could be taken into consideration certain change of title.

F. ex. " In vivo and in vitro models of obesity and pancreatic cancer, obesity - driven pancreatic cancerogenesis"

Round 2

Reviewer 2 Report

The manuscript is not improved. As mentioned earlier, no novelty and does not merit publication in the journal.